# Evaluation of convective boundary layer height estimates using radars operating at different frequency bands

Anna Franck[1], Dmitri Moisseev[1,2], Ville Vakkari[2,3], Matti Leskinen[1], Janne Lampilahti[1], Veli-Matti Kerminen[1], and Ewan O'Connor[2]

[1]Institute for Atmospheric and Earth System Research/Physics, Faculty of Science, University of Helsinki, Helsinki, Finland
[2]Finnish Meteorological Institute, Helsinki, Finland
[3]Atmospheric Chemistry Research Group, Chemical Resource Beneficiation, North-West University, Potchefstroom, South Africa

**Correspondence:** Anna Franck (anna.franck@helsinki.fi)

**Abstract.**

Knowledge of atmospheric boundary layer state and evolution is important for understanding air pollution and low level cloud development, among other things. There are a number of instruments and methods that are currently used to estimate boundary layer height (BLH). However, no single instrument is capable of providing BLH measurements in all weather condi- tions. We proposed a method to derive a daytime convective BLH using clear air echos in radar observations and investigated the consistency of these retrievals between different radar frequencies. We utilized data from three vertically-pointing radars that are available at the SMEAR II station in Finland: the C-band (5 GHz), Ka-band (35 GHz) and W-band (94 GHz). The Ka- or W- band cloud radars are an integral part of cloud profiling stations of pan-European Aerosol, Clouds and Trace Gases Research Infrastructure (ACTRIS). Our method will be utilized at ACTRIS stations to serve as an additional estimate of the BLH during summer months. During this period, insects and Bragg scatter are often responsible for clear air echos recorded by weather and cloud radars. To retrieve a BLH, we suggested a mechanism to separate passive and independently flying in- sects that works for all analyzed frequency bands. At the lower frequency (the C-band) insect scattering has been separated from Bragg scattering using a combination of radar reflectivity factor and linear depolarization ratio. Retrieved values of the BLH from all radars are in a good agreement when compared to the BLH obtained with the co-located Halo Doppler lidar and ERA5 reanalysis dataset. Our method showed some underestimation of the BLH after night-time heavy precipitation yet demonstrated a potential to serve as a reliable method to obtain a BLH during clear-sky days. Additionally, the entrainment zone was observed by the C-band radar above the CBL in the form of a Bragg scatter layer. Aircraft observations of vertical profiles of potential temperature and water vapour concentration, collected in the vicinity of the radar, demonstrated some agreement with the Bragg scatter layer.

## 1 Introduction

A convective boundary layer (CBL) often develops over land during the day when strong surface heating initiates buoyancy-driven turbulent convection. The turbulent motion efficiently mixes heat, moisture and other atmospheric constituents within

this layer. Since most atmospheric pollutants are emitted from sources at the surface, they are accumulated in the CBL, and the height and evolution of the CBL is essential for the monitoring and forecasting of air quality (Garratt, 1994).

The top of the convective layer is typically capped by a stable inversion layer that hinders further rise of the air (Stull, 1988). The height of the CBL top can be determined from the difference in the air properties, or air property tracers, in the boundary layer and above. Various direct and indirect measurement techniques have been developed to estimate the CBL height (CBLH) (Seibert et al., 2000). One of the most common direct observations of the CBLH is done using the vertical profiles of potential temperature and relative humidity measured by rawinsondes (Holzworth, 1967; Seidel et al., 2010). However, due to poor

spatial and temporal resolution of the rawinsonde network, other methods are needed for continuous monitoring of the CBLH. Continuous measurements of the atmosphere are provided by ground-based remote sensing instruments, which, among other instruments, include lidars and radars (Emeis et al., 2008). Lidars continuously measure aerosol vertical distribution, which is used to retrieve the CBLH with high temporal resolution (e.g., Baars et al., 2008). Different algorithms to diagnose the BLH using lidars are summarized in Dang et al. (2019). In relatively clean environments, like Finland, lidars' retrieval algorithms

encounter problems to estimate a correct CBLH due to low aerosol load, while in many other environments problems arise due to multiple aerosol layers (Granados-Muñoz et al., 2012). During recent years, aerosol lidars have been supplemented by Doppler lidars in BLH research (e.g., Vakkari et al., 2015; Hellén et al., 2018; Manninen et al., 2018). However, Doppler lidars are also limited by low signal in clean environments (Manninen et al., 2016).

    Scattering by small changes in the refractive index of the atmosphere, called Bragg scattering, can be detected by some

radar frequencies. These changes occur at the border of the CBL and free atmosphere due to large gradients in temperature and relative humidity. Starting from the middle of the last century many studies have shown that the S-band (10 GHz) radar is sensitive to fluctuations at the boundary layer top (e.g., Lhermitte, 1966; Hardy and Ottersten, 1969; Konrad, 1970). Heinselman et al. (2009) made the first assessment of a possibility to use the reflectivity factor of the USA Weather Surveillance Radar-1988 Doppler (WSR-88D) to obtain the depth of the CBL during clear sky and light winds. Elmore et al. (2012) proposed to add

solar measurements to improve the radar reflectivity-based method of Heinselman et al. (2009) and to automate the process with a linear regression model. After the US operational radar was upgraded with dual polarization, the differential reflectivity has been used to obtain the CBL top height for different locations across the USA (Richardson et al., 2017; Banghoff et al., 2018; Tanamachi et al., 2019). The main advantage of using the differential reflectivity is its capability to differentiate Bragg scatter from insects, birds and other nonspherical biota (Melnikov et al., 2011).

Biota signatures in the radar returns, on the other hand, can also help to obtain useful information about the development of the BL and its height. Very small insects (<10 mm in diameter) or "aerial plankton", as they have been also called by Drake and Farrow (1989), are of primary interest for the BL studies. These insects, such as aphids, are active during daytime and tend to be rather passive flyers as a means of conserving energy and use the turbulent updrafts and downdrafts for transport that they would otherwise not be able to undertake (Parry, 2013). Since the 1970s, it has been recognised that insects produce echos in

different radar frequencies (e.g., Atlas et al., 1970; Richter et al., 1973; Riley, 1985; Russell and Wilson, 1997; Contreras and Frasier, 2008; Clothiaux et al., 2000). For example, insect echoes in the scanning weather radars have been used to trace wind motions (Achtemeier, 1991; Wilson et al., 1994).

More recently, Chandra et al. (2010) made a long-term study of the daytime evolution of vertical velocity variance, mass flux and skewness in the CBL by utilizing insect echos in the vertically pointing Ka-band. Observations from the Ka- and W-bands were used by Wood et al. (2009) to compare the top altitude of insects with the CBL height obtained with a Vaisala CT75K ceilometer. They found that some insects often rise above the CBL top. Wainwright et al. (2017) studied the behaviour of small insects with the Ka-band radar and the Halo Doppler lidar and found, as similarly reported by Geerts and Miao (2005a, b), that these small daytime insects travel upwards with a slower pace than the airflow in the rising air plumes.

In recent years, research centers in Europe started to transform into research infrastructures and infrastructure networks (Bolliger and Griffiths, 2020). This transition requires harmonization of operational methods, data quality controls and delivered data products (Hirsikko et al., 2014). The cloud profiling stations of the pan-European Aerosol, Clouds and Trace Gases Research Infrastructure (ACTRIS, Pappalardo, 2018) typically operate either Ka- or W-bands cloud radars. In some cases, observations from these radars are supplemented by weather radar measurements that are operating as a part of the infrastructure or by national weather services (e. g. this is the case in Finland). The aim of this paper was to investigate a possibility of obtaining a CBLH during spring and summer months using insect echoes in radars operating at these frequencies and to compare the consistency of these retrievals; moreover, to evaluate the potential of Bragg scatter observed by the C-band radar to provide the CBLH and entrainment zone depth. These methods could serve as additional ways to estimate the CBLH following the recommendations of Seibert et al. (2000) and Emeis et al. (2008) to utilize several methods to obtain the true BLH.

A brief description of three cloud radars used in this article to obtain a CBLH is given in Sect. 2, as well as short introduction to Halo Doppler Lidar and ERA5 Reanalyses dataset that were used for evaluation of the results. A short theoretical basis to Bragg and insect scattering in Sect. 3 highlights the difference between the two scattering mechanisms and explains how they are used in our method of estimating CBLH. The description of the process of separating small daytime insects from larger actively flying daytime and night time insects and estimating a CBLH using these small insects as tracers (Sect. 3) is followed by several detailed example cases and evaluation of the method with monthly data (Sect. 4).

## 2 Measurement location and instrumentation

A radar field is located in Hyytiälä, southern Finland, at the Station for Measuring Forest Ecosystem–Atmosphere Relations (SMEAR II, 61°51′N, 24°17′E, 180 m above sea level, Hari and Kulmala, 2005). It is a rural background station representing a boreal forest environment that is not affected by anthropogenic emission sources. Aphids (order Hemiptera), flies (order Diptera), thrips (order Thysanoptera), wasps and ants (order Hymenoptera), butterflies and moths (order Lepidoptera) are common insects in the area, and some mass migrations of aphids in Finland have been described (Nieminen et al., 2000) and successfully forecast (Leskinen et al., 2011).

### 2.1 Cloud radars

Since November 2017, the W-band cloud radar (HYytiälä Doppler RAdar, HYDRA-W) is operating at the station as a part of ACTRIS. The HYDRA-W is a FM-CW system (Küchler et al., 2017). The lowest measurement height is 100 m above the radar

and the range resolution is about 25 m for the heights lower than 3600 m. In addition to the radar reflectivity, mean Doppler velocity, and spectral width, the radar measures the linear depolarization ratio (LDR). LDR is measured when a radar receives signal simultaneously in the horizontal and vertical polarization channels, and it helps to characterize shapes of the observed targets.

The C-band Hyytiälä Doppler Radar (HYDRA-C) has been continuously operating at the station since September 2016. It

is a dual-polarization weather radar that is currently operating in a vertically pointing mode. The radar uses a 0.5 ms pulse and the effective radar resolution, after averaging, is 100 m. In the current operation mode HYDRA-C also performs LDR measurements in addition to the standard radar spectral moments. For the comparison of the HYDRA-W and HYDRA-C observations see for example Li and Moisseev (2020). The temporal resolutions of HYDRA-C and HYDRA-W measurements are 1.37 and 3.35 s, respectively.

The Finnish Meteorological Institute deploys their scanning Ka-band Doppler cloud radar (MIRA-35, Görsdorf et al., 2011) to Hyytiälä for measurement campaign purposes. The radar is equipped with a 35 GHz magnetron transmitter and allows for 30 m range resolution. The temporal resolution of the measurements is 0.75 s. Besides reflectivity factor, Doppler velocity and Doppler spectrum width, this cloud radar also provides LDR measurements.

## 2.2 Halo Doppler lidar

A Halo Photonics Stream Line scanning Doppler lidar is operating continuously at Hyytiälä approx. 180 m northeast of the radar field. Halo Stream Line is a 1.565 $\mu$m pulsed Doppler lidar with a heterodyne detector, 90 m minimum range and 30 m range resolution (Pearson et al., 2009). At Hyytiälä, the Doppler lidar was configured to operate a 30° elevation angle vertical azimuth display (VAD) scan and a set of 12 beams of vertical stare with 40 s integration time every 30 minutes. Other measurements during the 30-min measurement cycle were not utilized here. The long integration time was used to overcome

the generally low signal levels at Hyytiälä (Manninen et al., 2016) and the data were post-processed according to Vakkari et al. (2019) to further reduce the noise floor. More detailed operating specifications of the lidar can be found in Hellén et al. (2018).

The VAD scan was used to retrieve the horizontal wind profile and a proxy for turbulent mixing, $\sigma^2_{\mathrm{VAD}}$, as described in Vakkari et al. (2015). The wind profile and vertically-pointing measurements were used to retrieve turbulent kinetic energy (TKE) dissipation rate according to O'Connor et al. (2010). Mixing layer height (MLH) was determined from these parameters

similar to Vakkari et al. (2015) and Hellén et al. (2018). In short, if TKE dissipation rate at the first usable range gate at 105 m (the heights are given above ground level everywhere in the text except when specified otherwise) was larger than $10^{-4}$ m$^2$ s$^{-3}$, MLH was taken as the last range gate where TKE dissipation rate was higher than this threshold. Otherwise, $\sigma^2_{\mathrm{VAD}}$ profile was used to identify turbulent mixing below 105 m. A threshold of $0.05$ m$^2$ s$^{-2}$ was used for $\sigma^2_{\mathrm{VAD}}$ to determine MLH below 105 m. With this combination MLH could be determined with 30 min resolution from 60 m to >2000 m, excluding rainy periods.

In this study, we will use Doppler lidar MLH for comparison with the CBLH derived from the radar echoes during daytime.

## 2.3 ERA5 Reanalysis data

ERA5 is a reanalysis database containing hourly estimates of atmospheric variables (Hersbach and Dee, 2016). They are calculated on a 30 km grid with 139 pressure levels approximately up to 80 km using an advanced four-dimensional variational assimilation scheme in the European Center for Medium-Range Weather Forecasts' System. We have used a BLH parameter, which is calculated based on an entraining parcel model for turbulent situations, and the bulk Richardson number algorithm (Vogelezang and Holtslag, 1996) for stable conditions.

## 2.4 Airborne in situ measurements

Potential temperature, relative humidity and temperature profiles were measured in the lower atmosphere on board a light Cessna 172 aircraft. An instrument rack was installed inside the airplane's cabin. A temperature and relative humidity sensor (Rotronic HygroClip-S) was mounted under the right wing. A pressure sensor (Vaisala PTB100B) was installed inside the unpressurized cabin. The measurement flight profiles were flown at the air speed of about 130 km/h and within a 40 km radius of Hyytiälä. A typical profile consisted of a steady ascent and descent, both lasting about 1 hour, with horizontally straight flight legs that were perpendicular to the mean wind direction. The maximum altitude was about 3800 m above sea level. More details about the set up and measurements can be found in Schobesberger et al. (2013), Leino et al. (2019), and Lampilahti et al. (2020).

## 3 Method

### 3.1 Theoretical basis

There are several approaches to estimate BLH using radar observations. For cm-wavelength radar (e.g. the C-band) Bragg scattering can be used. The Bragg scatter occurs in areas where there are strong perturbations in the refractive index of the atmosphere at scales on the order of half the radar wavelength. It can happen at the boundary between the CBL and the free troposphere where there is turbulent mixing across large gradients of temperature and humidity (Melnikov et al., 2011). This mixing leads to eddies with a range of sizes, some of which are resonant with the scale producing Bragg scatter (2.5 cm for the C-band radar). Heinselman et al. (2009), Richardson et al. (2017), Banghoff et al. (2018), and Tanamachi et al. (2019) used this approach to estimate BLH using S-band (10 cm) weather radar observations.

For mm-wavelength radars (e.g. the Ka- and W-bands) Bragg scattering is very small and cannot be observed. In such cases, reflections from scatterers that are passive tracers of air motion should be used. Insects that are present at most if not all cloud remote sensing sites during spring and summer months can act as such scatterers (Geerts and Miao, 2005b; Luke et al., 2008; Wood et al., 2009; Chandra et al., 2010). Small insects are often present in the amounts sufficient to act as volume radar targets. Given that their size may become comparable to the cloud radar wavelength, especially at W-band, a wavelength dependence of the radar echoes is expected. Larger, free flying, insects tend to cause point like returns in the radar observations. This difference between larger and smaller insects can be used for their discrimination.

In order to separate radar insect returns from clouds, dual-polarization radar observations such as LDR can be used (Martner and Moran, 2001). Given that insects are present in spring and summer and they are seldom found at temperatures below 0 C, it is important to separate radar echoes of water cloud and drizzle from these of insects. Given that cloud and drizzle droplets are almost spherical, their LDR would be very small. The LDR of insects, on the other hand, is expected to be high.

## 3.2 Process of estimating CBLH during a clear-sky day

Figure 1 shows how a CBL development is seen during clear-sky conditions in the reflectivity factors (Ze) from all radars present at the SMEAR II station on 9 May 2018: the C-band (Fig. 1a), Ka-band (Fig. 1c) and W-band (Fig. 1d). During this day, lots of insects were present in the low atmosphere mostly below 1300 m as seen from the radars. The C-band is sensitive mostly to insects during the day due to their large amount, the W-band also gets most of the signal from the numerous day time insects as well as some signal from the independent night-time insects, whereas the Ka-band is the most sensitive to the night-time insects. Besides insects, the C-band is also capable of detecting Bragg scatter, which can be visually distinguished from the insects: Bragg scatter appears like a continuous line above insects with quite similar reflectivity (Fig. 1a) and different LDR values (Fig. 1b). Insects have nonspherical shapes with LDR values closer to zero, while Bragg scatter, on the other hand, has low depolarization ratio, less than -20 dB.

In the morning between about 08.00 and 10.00 UTC, the CBL development process is mainly seen in the C-band, showing that the CBLH increased from about 800 m to 1200 m during this time period. Some insects are seen in the Ka- and W-bands, but not as high up in the atmosphere. This might be due to the fact that earlier in the morning, the air near the surface is not warm enough to trigger strong updrafts, whereas the echos in the C-band can mostly be due to Bragg scatter at this time.

Starting from about 10.00 and continuing until about 16.00 UTC, the CBLH is visually seen from the top altitude of insects to be around 1200 m. Between about 16.00 and 18.00 UTC when the CBL starts to dissipate, there are still some insects seen in the Ka-band, but not as many in the C- and W-bands. Bragg scatter is also seen to remain a bit longer, until almost 17.30 UTC. In the evening, night-time insects started to appear in the Ka- and W-bands below 800 m. Those are independently flying insects and do not exactly follow the air motion, but can be helpful for the retrieval of the nocturnal boundary layer height (Wainwright et al., 2020).

## 3.3 Bragg and insects mask from the C-band radar

The first step in obtaining a BLH from the insect echoes in the C-band radar is to separate the insect from Bragg scattering. For this purpose, we try to utilize a combination of Ze (Fig. 1a) and LDR (Fig. 1b). Histograms on Fig. 2 show the distributions for insect and Bragg scattering. The histogram that displays distributions of Ze and LDR related to insects(Fig. 2a) was obtained from the measurements during 9 May for the altitude range 950 to 1100 m and for the time interval between 12.00 and 14.00 UTC. The peak in the LDR for insects is observed at -5 dB and the values cover the region between -33 and -3 dB. For the histogram of Bragg scatter, the altitude range was between 1200 and 1400 m at 10.00 to 12.00 UTC. From Fig. 2, we can see that most of the LDR values for Bragg scatter are located between -40 and -5 dB with a peak at -26 dB. The distribution of

values in the Bragg histogram is affected by insects present in the atmosphere at the same time and at the same height, but is impossible to avoid. This will be taken into account later in the process.

Based on Ze and LDR distributions, a Bragg-insect mask for the C-band was created following the steps in the schematic diagram presented in Fig. 3. Both the lowest and highest boundary values of Ze and LDR can be directly assigned as Bragg or insects pixels. There is a region in LDR (between -33 and -5 dB) and Ze (between -29 and -5 dBZ) distributions where both insects and Bragg values intersect and cannot be easily categorized. In order to assign these questionable pixels, firstly, pixels with values closest to LDR of Bragg scatterers (between -20 and -16 dB) are assigned as Bragg if they are surrounded by at least four Bragg pixels. If this condition is not fulfilled, they are moved to the second group of questionable pixels that consists of measurements with LDR values between -16 and -14 dB. Surroundings of these pixels are also checked: if at least five pixels around are Bragg pixels, then the pixel is classified as Bragg; if four or more are insects, then the pixel is classified as insects; if none of the conditions are satisfied, then the pixel remains uncategorized.

An obtained Bragg-insect mask for 9 May 2018 is shown in Fig. 4, where yellow colour represents insects and a more continuous black line illustrates Bragg scatter. We can also see some brown color pixels on the edges of Bragg scatter, that remained unclassified and can be either Bragg or insects. The height of the CBL can be derived using this Bragg-insect mask. The Bragg area corresponds to the entrainment zone, while both the lower boundary of the Bragg mask and the upper boundary of the insect mask indicate the CBLH.

## 3.4 Passive and independently-flying insects in the Ka- and W-bands

During clear-sky spring and summer days the radar returns of the Ka- and W-bands are predominantly composed of insects as seen in Fig. 1c and 1d. For the CBLH retrieval, we are interested in small passively moving insects that are mainly following the air motion in the CBL. In the radar reflectivity factors (Fig. 1b and 1d), besides those passively flying insects, there are also many insects seen in the evening and night at lower heights. To exclude these independently flying insects from the passive insects of our interest, we have calculated the standard deviation of radar velocities for three minutes intervals for the Ka-band (Fig. 5a) and the W-band (Fig. 5b). The contrast in the day-time and evening-time values is seen quite well. Based on a visual inspection, we set a value of 0.5 ms$^{-1}$ as a threshold for passive insects: if values are lower than 0.5 ms$^{-1}$, then we assume that there are independently flying insects that could not be used for our method. This is especially important for separating afternoon transition from the unstable to stable boundary layer. We also have also tried to look at a spectral width, but the discrepancies are not seen there as clearly.

## 4   Results and Discussion

### 4.1   The CBLH from all radars

Figure 6 shows the retrieved CBLH using the Bragg-insect mask from the C-band (yellow) and insects filter from the Ka-band (purple) and the W-band (green) on 9 May 2018. There is a good agreement in the CBL derived from the insects echos from

all the radars during the daytime between about 10.00 and 17.00 UTC. Discrepancies arise in the morning from 06.00 to 09.00 UTC: the CBLH is a bit higher in the Ka-band (around 500 m) and lower in the W-band (300-400 m), whereas in the C-band the retrieval is only possible starting from about 09.00 UTC. These discrepancies in the morning appear due to different radar sensitivities to the small number of insects in the atmosphere.

The obtained CBLH is compared to the BLH derived from the Halo Doppler lidar and the ERA5 reanalysis data, shown in Fig. 6 using blue and red colours, respectively. The height of the CBL during the day is similar for all methods, except for the morning values again. The CBLH obtained with insects has approximately 90 min delay from the ERA5 and Halo lidar, probably due to the fact that there were no strong updrafts yet at this hour and, therefore, not many insects higher in the atmosphere. Moreover, it has been shown that insects prefer temperatures of higher than 10 ° C for comfort flying (Wilson et al., 1994; Drake and Reynolds, 2012), therefore, the lag between the CBLH derived from radars and that obtained using other methods might be due to the low temperature at the CBL top.

## 4.2 The entrainment zone

The Bragg scatter area (in grey, Fig. 6) is situated above the CBLH derived from the insects and has a depth of 100-250 m during the day. This area might represent the entrainment zone, a transition zone where air from above is mixed into the CBL contributing to the growth of the layer. It is seen that the area is a bit smaller in the morning, grows during the afternoon up to 250 m, and shrinks to 50 m in the evening before disappearing completely at 17.30 UTC. Interestingly, this zone is still present between about 16.30 and 17.30 UTC when all other methods show that the CBL has already started to dissipate.

One of the Cessna flights was performed in the afternoon (10.15 -12.45 UTC) on 9 May 2018. Figure 7 shows vertical profiles of the potential temperature, water vapour concentration (WV) and temperature during the descent, superimposed on the Bragg scatter area. Cessna reached the entrainment zone at 12.15 UTC, approximately 15 km away from the radar field. Changes in the temperatures and WV are clearly seen in all three graphs, when the profiles intersect with the Bragg scatter lower edge. The WV profile is quite constant inside the entrainment zone and above. The potential temperature and temperature, however, are decreasing with height for 70 meters until another change is seen in both profiles - very small in the temperature, yet obvious in the potential temperature. These changes take place a bit lower than the upper edge of the obtained Bragg scatter layer, which might be due to averaging of the radar data and distance between the radar field and Cessna location. Further analyses with more flight days are needed to make a conclusion on the depth of the entrainment zone.

## 4.3 Clouds and precipitation case

In the presence of low level clouds or precipitation, the BLH can not be obtained with our method. We tested our CBLH retrieval method for a clear-sky case after heavy precipitation during the night. As explained in Section 3, clouds and insects can be separated based on LDR: for insects LDR is expected to be high, while for more spherical cloud droplets and precipitation very small. Using this knowledge we created a cloud mask using Ze and LDR of the W-band radar. In the C-band, on the other hand, Ze and LDR of clouds and precipitation are very similar to Ze and LDR of Bragg scatter, therefore, it is impossible to separate them. To overcome this problem, we used the advantage of having several radars at the field, and applied cloud mask of the

W-band radar to the C-band data. The reflectivity factors from available radars and LDR from the C-band radar are shown for a case with night clouds and precipitation on 18 May 2018 (Fig. 8a-d). It can be seen that clouds were present during the whole night until 06.00 UTC and there were several rain events. The BL started to develop after the rain had passed, which can be seen in the radars' echoes from 500 m at 06.00 UTC. The BLH derived from the Ka-band, similarly to the first case, shows more insects and at a bit higher altitude compared to other radars.

Figure 9a presents the C-band Bragg-insect mask together with the clouds/precipitation mask, that was created using the W-band data. We can notice that the Bragg scatter zone is quite extensive and reach up to 2500 m in the afternoon, while there are not as many insects during the day with quite big gaps. There is also an interesting part of Bragg scatter in between precipitation at 04.00 UTC.

The obtained CBLH profiles retrieved for this day do not coincide with each other as well as for the first case (Fig. 9b). The profiles of the C-band and W-band have the most similarities between each other. Compared to the Halo Doppler lidar and ERA5, the CBLH derived from the Ka-band is the closest to them from 6.00 to 13.00 UTC, but with up to 250 m under-estimation in the afternoon. The CBLH profiles from the C-band and the W-band insects are up to 800 m lower than Doppler lidar during that time. Underestimation of the CBLH from the C-band data can be explained by the lower sensitivity of this wavelength to these small insects, while underestimation of the W-band data could be related to flaws in the algorithm, that might have identified passively flying insects as independent because there were not as many of them higher up (Fig. 8d) In the evening, the starting point of the CBL dissipation agreed quite well between our methods and Halo Doppler lidar, to be around 16.30 UTC, while ERA5 showed a bit different transitioning profile.

The entrainment zone obtained from the Bragg scatter follows more closely the ERA5 and Halo Doppler lidar. From 06.00 until 13.00 UTC, its upper edge follows the ERA5 estimates and the lower edge is at the same heights as the Doppler lidar retrievals, and in the afternoon the entrainment zone is mostly above both of them. Unfortunately, Cessna flights were not conducted during this day and we were not able to verify the entrainment zone with temperature and humidity profiles.

## 4.4   Evaluation of the obtained CBLH

We used our algorithm to calculate the CBLH for May 2018 for the C- and W-bands. The Ka-band did not work during the whole time-period, so there were only eight days of measurements, some of which we chose for the case studies. Hourly mean values of the CBLH obtained using insects echos in the W-band (Fig. 10) and the C-band (Fig. 11) are compared with hourly mean Halo Doppler lidar measurements and ERA5. Derived CBLH profiles from the W-band radar follow more closely the Doppler lidar observations of the BLH during most of the days, yet in a good agreement with ERA5 also. Days with low level clouds and precipitation, such as 16 and 17 May, are missing from the W-band retrievals. There is also an agreement in the methods of the starting time of the CBL development in the morning and transitioning time in the evening.

Larger discrepancies are observed when comparing CBLH obtained with the insects echos in the C-band radar with Doppler lidar and ERA5 products (Fig. 11). In most cases the retrieved CBLH was slightly lower than the ERA5 and Halo Doppler lidar CBLH estimates. The reason might be due to the lower sensitivity of the C-band to the smaller amounts of insects that rise to the top of the CBL. Nevertheless, the overall performance of the method for the C-band radar is comparable to that of

the ERA5 and Halo Doppler lidar CBL estimates: the typical CBL profiles are seen every day in May with growth and decay times matching the CBLH obtained with other two methods.

A direct comparisons of the obtained CBLH from the W-, Ka- and C-band radars with Halo Doppler lidar and a bivariate fit are shown in Fig. 12, where data from six available days are used from the Ka-band radar. Colors represent the hour of the day and are chosen to be from 06.00 to 15.00 UTC as a typical time for the CBL to evolve during the day. The scatter plot between the CBLH obtained using insects in the millimeter wavelength radars (the Ka-band and W-band) and the CBLH from the Halo Doppler lidar (Fig. 12c) shows the better agreement between parameters (R=0.9 and R=0.84, respectively), while for the C-band (Fig. 12b) R is 0.76. The comparison plots highlights that the most discrepancies occur during the afternoon transition period, when there are still some passively-flying insects present in the lower atmosphere. On the graphs it can be seen in the top left quadrant, where Halo Doppler lidar CBLH is lower than 1000 m, while our algorithm shows heights of around 1500 m.

## 5  Conclusions

We have proposed a method to derive a daytime CBLH using clear air echos in the radar observations and investigated the consistency of these retrievals between different radar frequencies. We have utilized data from three radars that are available at the SMEAR II station in Finland: the C-band (5 GHz), Ka-band(35 GHz) and W-band(94 GHz). The Ka- or W- band cloud radars are an integral part of cloud profiling stations of pan-European Aerosol, Clouds and Trace Gases Research Infrastructure (ACTRIS). After validating our method with data from different geographical locations, it will be utilized at the ACTRIS stations to serve as an additional method of obtaining the CBLH during spring and summer months.

Insects and Bragg scatter are often responsible for the clear air echos recorded by the radars. These echoes are used in this study to estimate the CBLH. The main challenge in using insect echoes for CBLH retrieval is to use only small insects that follow the air motion. We have found a mechanism to separate these small weakly flying insects from the independently-flying ones based on Doppler velocity that can be applied to all used radar frequency bands. The obtained CBLH using insects was compared to the CBLH retrieved from a co-located Halo Doppler lidar and ERA5 reanalysis dataset for May 2018. All CBLH profiles follow each other quite close during the clear sky days. The agreement between the CBLH from the Halo Doppler lidar and the retrieved CBLH from the W-band radar(R=0.84) is a bit better than between Halo Doppler lidar CBLH and the CBLH retrieved from the C-band radar (R=0.76). The most difficult time of the day to derive a CBL top is the CBL transition period from unstable to stable in the late afternoon, as some small passive insects can stay higher in the atmosphere, even when updrafts already weakened (Wainwright et al., 2020). Another difficulty appears during morning hours, where in some cases a time lag was identified between the CBLH derived from the radars and other methods, which might be due to the temperature threshold for insect flight.

One case study has shown that on the day following heavy precipitation during the night, the amount of insects in the CBL is not sufficient for the radar to obtain the correct CBLH. The retrievals from the Ka-band was up to 300 m lower and around 800 m lower for the C-band and W-band compared to the Halo Doppler lidar and ERA5. The short dataset did not provide enough

information to make conclusions about the possible time when precipitation would have to cease in order to be able to obtain the CBLH with the proposed method.

Besides insect echoes, the C-band radar also detects Bragg scatter that appears due to small changes in the refractive index of the atmosphere. We have separated Bragg from the insect scattering based on the LDR and Ze. The lower edge of the Bragg scatter area corresponds to the CBLH. The whole Bragg scatter area obtained from the C-band might represent the

entrainment zone. The lower edge of the Bragg scatter zone matched well with the changes in potential temperature, water vapour concentration and temperature profiles from the airborne in situ measurements, while the upper edge did not match so well and was approximately 20 meters higher. The knowledge of the depth and location of the entrainment zone can help in estimating entrainment rate and, thus, the forecasting of the CBL development. However, more data are needed for further analyses of the depth of the entrainment zone.

To conclude, a CBLH retrieved using insect echoes from the millimeter wavelength radars such as the Ka-band and W-band could serve as a reliable method after being validated in other geographical locations, where different insect flight behaviour can be observed. From the cm-wavelength radar, such as the C-band, a CBLH can also be retrieved using insect echoes, however, the sensitivity to insects of this type of radar is lower and the obtained values of CBLH can be underestimated.

*Data availability.* The presented HYDRA-W and HYDRA-C data are available by request from DM. The Ka-band data are available by

request from EOC and Halo Doppler lidar MLH from VV. Airborne data can be found from https://doi.org/10.5281/zenodo.4063662. The ERA5 dataset was downloaded from https://cds.climate.copernicus.eu and chosen for the grid point closest to Hyytiälä station.

*Author contributions.* DM, EOC and AF conceptualized the study. DM initiated radar observations in Hyytiälä and pre-processed HYDRA-C and HYDRA-W data. EOC pre-prosessed the Ka-band radar and Halo Doppler lidar data. VV analysed Halo Doppler lidar data. ML supervised the radar observations on the radar field and contributed with his expertise of insects echos in radar returns. JL organized and

coordinated Cessna flight campaign. AF performed data analysis and constructed the figures. AF has planned the manuscript, and all authors contributed to writing and reviewing it.

*Competing interests.* The authors declare no competing interests.

*Acknowledgements.* This research has been supported by the Horizon 2020 (grant nos. ACTRIS-2 654109, ACTRIS PPP 739530, ACTRIS-IMP 871115) and Academy of Finland (ACTRIS-NF 328616, ACTRIS-CF 329274, and Center of Excellence in Atmospheric Sciences,

307331, Atmosphere and Climate Competence Center, 337549), University of Helsinki (ACTRIS-HY). The authors would like to gratefully acknowledge Dr. Victoria Sinclair for her expertise in working with ERA5 reanalysis data.

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

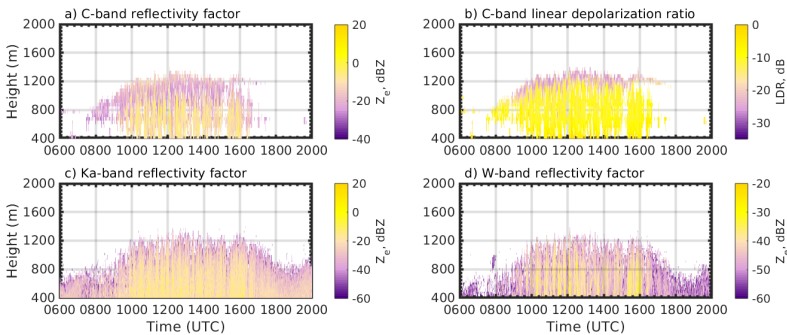

**Figure 1.** One minute averaged radar returns for 9 May 2018 in Hyytiälä: a) the C-band reflectivity factor, b) the C-band linear depolarization ratio, c) the Ka-band reflectivity factor and d)the W-band reflectivity factor. The CBL development can be visually seen from the radar returns. Note different colorbar scale limits in the graphs.

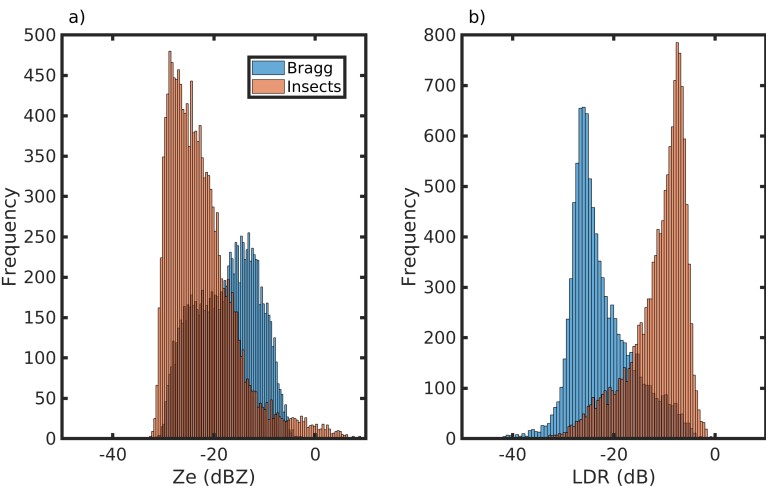

**Figure 2.** Histograms that show distribution of values of a) Ze and b) LDR for Bragg scatter(blue) and insects(red) during 9 May 2018. Bragg scatter and insects can be easier separated with LDR, which shows higher values for insects because of their non-spherical shape.

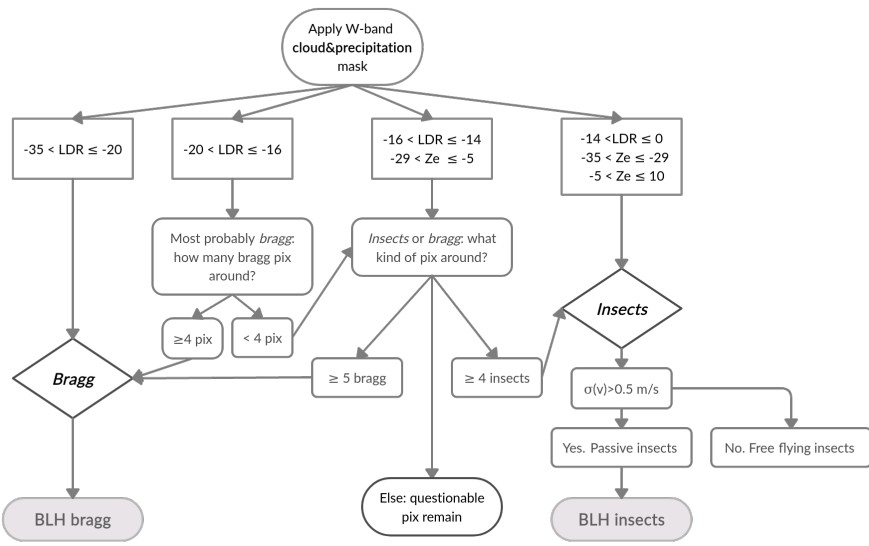

**Figure 3.** Schematic illustration of a decision tree based on Ze and LDR that is used to assign measurement pixels and to create a Bragg-insects classification mask.

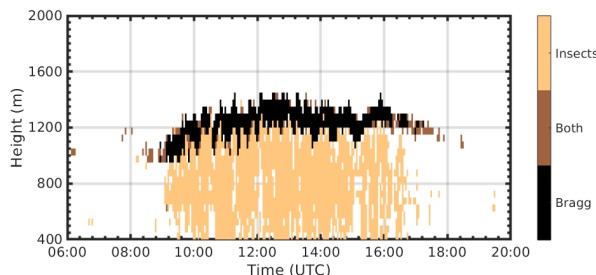

**Figure 4.** The Bragg-insects classification mask for 9 May 2018. Bragg scatter is shown with a continuous black line in the upper part of the CBL and insects (in yellow) occupy most of the CBL. Measurements that remain unclassified are coloured in brown.

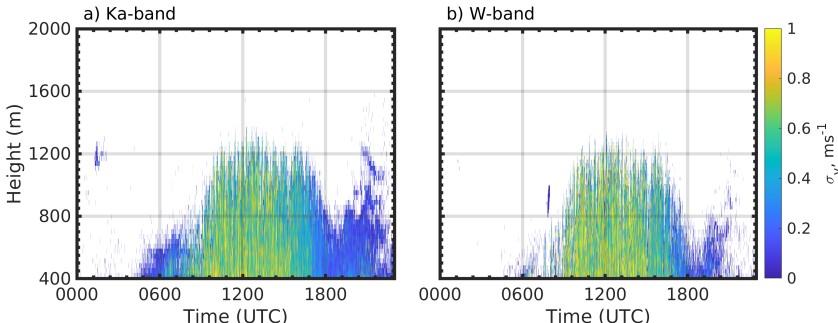

**Figure 5.** One standard deviation of a mean vertical velocity for the a) Ka-band and b) W-band radars for 9 May 2018. Lower values (<0.5 ms$^{-1}$) suggest actively flying insects, while high values indicate small passive flying insects.

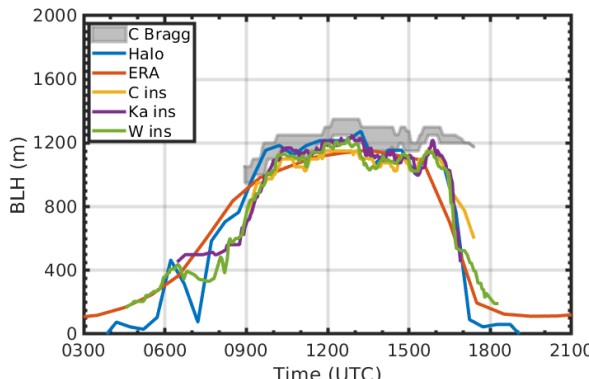

**Figure 6.** The CBLH derived from available radars compared to the CBLH retrieved from Halo Doppler lidar and ERA5 during 9 May 2018. All methods show good agreement during afternoon but have discrepancies in the morning and evening.

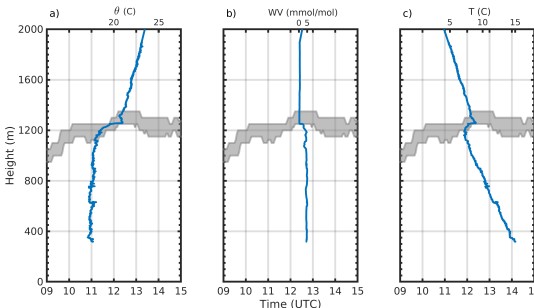

**Figure 7.** Profiles of the a) potential temperature, b) water vapour concentration and c) temperature during a Cessna flight superimposed on the Bragg scatter area, derived from the C-band radar on 9 May 2018. Overpass time for Cessna at 1250m is 12.15 UTC. Note, the x-scales for the flight variables are on top of the graphs.

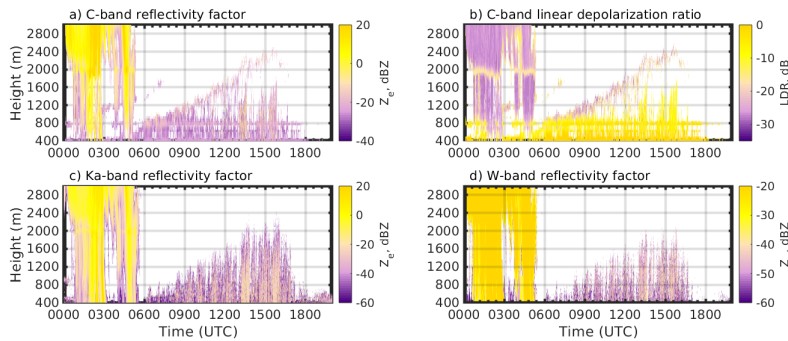

**Figure 8.** One minute averaged radar returns for 18 May 2018 in Hyytiälä: a) the C-band reflectivity factor, b) the C-band linear depolarization ratio, c) the Ka-band reflectivity factor and d) the W-band reflectivity factor. Clouds and precipitation are seen during the night until 06.00 UTC.

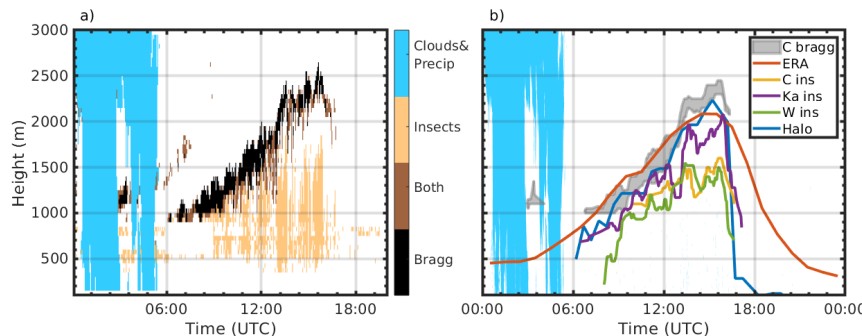

**Figure 9.** a)The Bragg-insects classification mask derived from the C-band radar combined with a cloud/precipitation mask from the W-band radar for 18 May 2018. b)Derived CBLH from all available radars compared with Halo Doppler lidar and ERA5.

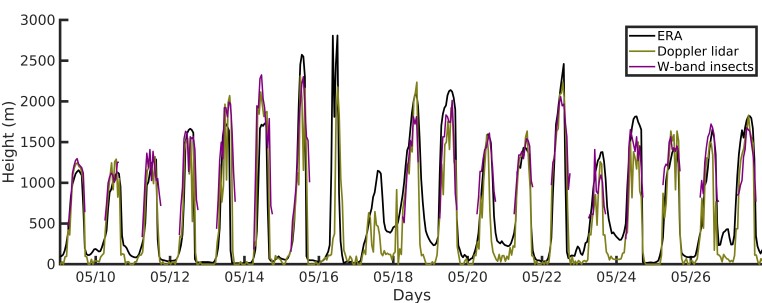

**Figure 10.** Time series of the CBLH during May 2018 derived using insects from the W-band radar compared to the Halo Doppler lidar retrievals and ERA5.

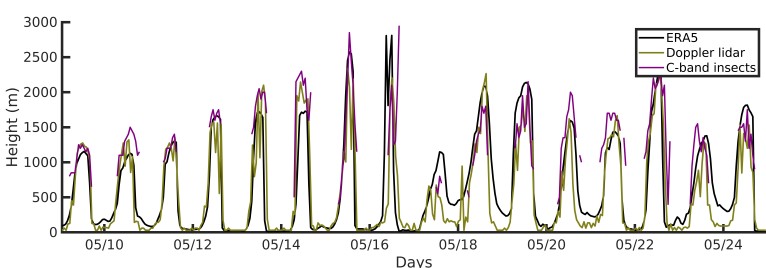

**Figure 11.** Time series of the CBLH during May 2018 derived using insects from the C-band radar compared to the Halo Doppler lidar retrievals and ERA5.

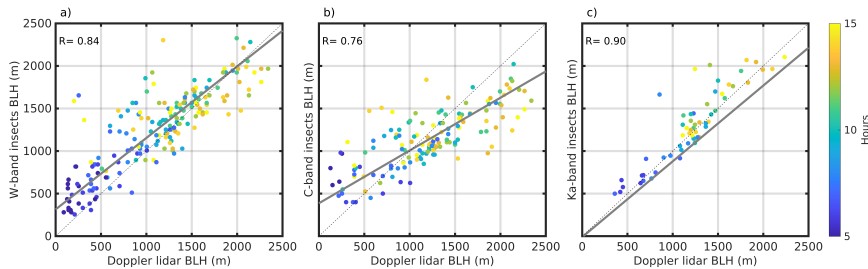

**Figure 12.** A comparison of the CBLH derived from the Halo Doppler lidar and a) the W-band, b) the C-band and c) the Ka-band radars. Colors represent different hours of the day. Note, that only six days of observations were used for the comparison of the Ka-band and the Doppler lidar.