# Peer review of "Evaluation of convective boundary layer height estimates using radars operating at different frequency bands"

_Atmospheric Measurement Techniques, 2021_

## Author Comment (AC1)

**Author's reply to the Anonymous Referee #1**

We thank the Referee for the constructive comments to help us to improve the manuscript. Below please find our answers to the comments.

Overview

In this manuscript the authors present a method for determining boundary-layer height from clear-air radar returns during the summer months. They test this method using data from collocated vertically pointing cloud radars at three different frequencies in Hytiälä, FInland and compare the resulting BLH values between the three radars as well as with ERA reanalysis and lidar-derived boundary-layer heights.

**General comments**

This paper is interesting and well written. It is great to see more researchers making use of the insect 'clutter' on cloud radars. The method described in the manuscript is robust and should provide a good platform for future studies in other areas or over a more extended time period. The figures are clear and helpful to the reader.

I have only minor comments and a few technical corrections that should be addressed prior to publication.

**Minor comments**

1. L147: Suggest also including Luke et al. (2008) here [Luke, E. P., Kollias, P., Johnson, K. L. and Clothiaux, E.E.(2008) A technique for the automatic detection of insect clutter in cloud radar returns. Journal of Atmospheric and Oceanic Technology, 25, 1498–1513]

   *We have added the reference as suggested by the Referee.*

2. In the schematic shown in Fig. 3, following in from the 'how many Bragg pix around' box, the options are >4 or <4. What happens if there are =4 Bragg pixels surrounding the pixel in question? Or should one of the options in the figure read ≥4 or ≤4? Similarly, the situation when LDR = -14 or Ze = -5 are undefined, so maybe there should be a ≤ or ≥ in those boxes somewhere too?

*The schematic was corrected as noted by the Referee. In the code the algorithm was correct. Also there was supposed to be '-' in front of 20 in the second to the left box, which was also corrected.*

[Figure]

Figure 1: Corrected infographic.

3. How was the 0.5 m/s threshold between passively or actively flying insects determined? Was this done visually based on Fig. 5, and were other threshold values tested to see how much difference this value made?

*Yes, it was done visually, taking into account the profile of ERA5 reanalysis BLH as seen in the attached figure. Text was changed to include that it was done visually: "**Based on a visual inspection,** we set a value of 0.5 as a threshold for passive insects:.."*

[Figure]

Figure 2: Comparison of sigma from the W-band radar and ERA5 re-analysis.

4. Since it is generally thought that the threshold temperature for insect flight is closer to 10 degrees C (e.g., Drake and Reynolds 2012), how does this impact your data? Do you think that part of the cause of the method not working as well during the morning transition might be due to the temperature at the CBL top being lower than 10 degrees C?

*Temperature definitely plays an important role in insects flight patterns. We have added text to the section 4 (line 222):* **"Moreover, it has been shown that insects prefer temperatures of higher than $10°C$ for comfort flying (Wilson et al, 1994; Drake and Reynolds, 2012), therefore, the lag between the CBLH derived from radars and that obtained using other methods might be due to the low temperature at the CBL top."**

*And also we have added to the conclusions to highlight the potential for future improvements (line 301):* **"Another difficulty appears during morning hours, where in some cases the time lag was identified between the CBLH derived from the radars and other methods, which might be due to the temperature threshold of comfort insects flight."**

5. You mention that Wood et al. (2009) find that insects are sometimes present at heights beyond the CBL top. This effect is also visible in Banghoff et al. (2018) and Contreras and Frasier (2008). How would you expect the presence of insects above the CBL to impact the performance of your algorithm? This seems to more commonly occur in regions with high temperatures and I appreciate that it may not have occurred in your dataset, but it is an important consideration for researchers who would like to apply this method in regions with very high summertime temperatures.

*Some insects present higher in the BL are independently flying insects, that should be identified by the sigma threshold. If these are passively flying insects, then some more testing of the algorithm should be done and may be some additional parameters, for example spectral width together with sigma. However, it is hard to speculate without proper dataset and the next step would be to test our method in different geographical locations. We have added to the conclusion (line 290):* **After validating our method with data from different geographical locations, it will be utilized at the ACTRIS stations.**

**Technical corrections**
*Technical corrections were implemented according to suggestions:*
L37-38: "Doppler lidars are also limited"

L40: "due to large gradients"

L52: "are of main interest"

L54: "Since the 1970s"

L56: "insect echoes"

L58: "More recently, Chandra et al. (2010)"

L64: "In recent years"

L70: "insect echoes"

L71: "compare the consistency"

L71: "observed by the C-band radar"

L95: "The radar uses a 0.5 ms pulse"

L104: "this cloud radar also provides LDR measurements"

L132: "steady ascent and descent"

L145: "For mm-wavelength radars"

L146: "to act as volume radar targets"

L176: "from the insect echoes"

L195: "is shown in Fig. 4"

L206: "we set a value of 0.5" - please add units here

L230: "during the descent"

L307: "corresponds to the CBLH"

L281: Should these values of R be $R^2$?

*We have calculated values of R, so it is correct in the text.*

---

## Author Comment (AC2)

**Authors reply to the Anonymous Referee# 2**

We thank the the Anonymous Referee #2 for the constructive comments to help us to improve the manuscript. Below please find our answers to the comments.

**General Comments**

The authors present novel methods for deriving boundary layer height (BLH) estimates from C-, Ka-, and W-band radars using a variety of filtering/masking techniques to parse out Bragg scatter and passive/active insects. These data are then compared alongside ERA5 and Halo Doppler lidar data to draw conclusions about the effectiveness of each radar in estimating BLH. The authors conclude that it is possible to derive BLH estimates from all 3 radar types, though each has its limitations. This manuscript expands upon previous findings in the realm of BLH estimation from radar and provides a beneficial framework for future research, which can hopefully provide a larger dataset and even more robust analysis to determine which radars are most effective at estimating BLH. The authors are to be commended for a scientifically-sound manuscript that is well-organized and easy to understand.

**Specific Comments**

1. The case studies for 9 May and 18 May provide useful context for the reader to understand the differences between various BLH estimation methods. Given that Figures 10 and 11 include 19 days' worth of data, and the paper lists that there are at least 6 other days for which Ka-band data exists outside of those case studies, it would be helpful to include some statistical information/analysis about the performance of each radar relative to the ERA5 and Doppler lidar (if applicable). Figure 12 provides helpful context comparing lidar to W- and C-band, and similar analysis would be beneficial with Ka-band and the ERA5 data. Perhaps you could consider a standard deviation time series throughout the day with ERA5 analysis as the baseline. Presumably this would visually display the errors inherent during the transition period in the afternoon/evening, among other things. Such a statistical analysis would add substantial value to this paper and make the results more robust.

   *A plot that shows a comparison between the Ka-band BLH and Doppler lidar with six days of data was added to Figure 12 (which shows similar comparisons between the W-*

*and C-band with the Doppler lidar). Also, the text was modified with the information about the added plot:* **"A direct comparisons of the obtained CBLH from the W-, Ka- and C-band radars with Halo Doppler radar and a bivariate fit are shown in Fig. 12, where data from six available days are used from the K-band radar. Colors represent the hour of the day and is chosen to be from 06.00 to 15.00 UTC as a typical time for the CBL to evolve during the day. The scatter plot between the CBLH obtained using insects in the millimeter wavelengths radars (the Ka-band and W-band) and the CBLH from the Halo Doppler lidar (Fig. 12c) shows the better agreement between parameters (R=0.9 and R=0.84, respectively), while for the C-band (Fig. 12b) R is 0.76."**

[Figure]

Figure 1: A comparison of the CBLH derived from the Ka-band radar with Halo Doppler lidar where colors represent different hours of the day

2. One of the main concerns with all three radars appears to be the lag time between the ERA5 BLH rise and the successive increase in radar-estimated BLHs. This appears to be a significant finding as well, especially as it relates to past studies that have found radar-estimated BLHs to be somewhat useful, in combination with other data sources, for assimilating into models and helping with convection initiation. At the very least, this lag time should be highlighted more in the text. I think this could also be added to the conclusion in a paragraph about future direction. Do you have any thoughts on how to mitigate this lag time, which is almost certainly important for forecasting? Could additional case studies and more robust dataset identify a "climatology" for BLH rise start time under given conditions? I think there are several different ways this discussion

could go, and you are certainly the experts on the state of using radar to estimate BLH at this time!

*At this first stage of introducing our method, it is difficult to make suggestions how to mitigate this lag problem, as it should be studied in other places and also with larger dataset. It might be related to the insects dependence on the temperature as suggested by the anonymous referee #1, which is certainly a good starting point for future work on this problem. To highlight this problem in the text more and to give some future directions, we have added text to the conclusion(line 301):* **"Another difficulty appears during morning hours, where in some cases the time lag was identified between the CBLH derived from the radars and other methods, that might be due to temperature threshold of comfort insects flight."**

And also we have added text to the section 4 (line 222): **"Moreover, it has been shown that insects prefer temperatures of higher than 10°C for comfort flying (Wilson et al, 1994; Drake and Reynolds, 2012), therefore, the lag between CBLH derived from radars and other methods might be due to the low temperature at the CBL top."**

3. Another interesting point you present is that CBLH retrieval following heavy precipitation overnight is difficult because the amount of insects in the CBL is not sufficient for the radar to obtain the correct CBLH. Is it possible that there is a certain time of day or night by which precipitation needs to cease in order for the CBLH to be estimated? Can you deduce anything from the other cases you sampled? At the very least, it seems like something that could be deduced from future work. This discussion would fit well at the end of line 304. Could you also surmise about why the C- and W-band radars were so much lower than the Ka-band? This conversation leads well into the next point...

*This was quite an exceptional case in our dataset that included only one month of data. Therefore, this can be open questions for the future work and we also added some text in the conclusion (line 304):* **"The short dataset did not provide enough information to make conclusions about the possible time when precipitation have to cease in order to be able to obtain the CBLH with the proposed method."**

*We have also added to Section 4.3 (line 259):* **Underestimation of the C-band can be explained by the lower sensitivity of this wavelength to these small insects, while underestimation of the W-band could be related to flaws of the algorithm, that might have identified passively flying insects as independent because there were not as many of them higher up (Fig.8d).**

4. With a manuscript like this that does such a good job of comparing the various methods for determining BLH, I was left wanting a concise explanation for the pros and cons of each method for estimating BLH. Whether this discussion is included in Section 4.4 in a more concise format or added to Section 5, a paragraph laying out the strengths and weaknesses of each radar would be prudent. Such a conclusion can likely be made from this study, subject to modification as additional cases are explored.

*We have added to the conclusion (line 312):* **To conclude, a CBLH retrieved using insects echoes from the millimeter wavelength radars such as the Ka-band and W-band could serve as a reliable method after being validated in other geographical locations, where different insects flying behaviour can be observed. From the cm-wavelength radar, such as the C-band, a CBLH can also be retrieved using insects echoes, however, the sensitivity to insects of this type of radars is lower and the obtained values of CBLH can be underestimated.**

**Technical Corrections**

*Technical corrections were implemented according to suggestions*:

Line 17 : . . . in the form of a Bragg scatter layer.

Line 32: . . . vertical distribution, which is used. . .

Line 37-38: . . . However, Doppler lidars are also limited by. . .

Line 52: . . . are of primary interest for BL studies. . .

Line 53: . . . passive flyers as a means of conserving energy. . .

Line 54: Since the 1970s. . .

Line 58: More recently, Chandra et al. (2010) made. . .

Like 69: ...services, (e.g. this is the case in Finland).

Line 71: ....retrievals; moreover, to....

Line 75: ..Reanalyses dataset that. . .

Line 103: ...also provides LDR measurements.

Line 112: ...used to retrieve the horizontal wind profile...

Line 118: ...was used for VAD2to determine the MLH

Lines 124-125: We have used a BLH "parameter that" or "parameter, which"....

Was this BLH parameter one of many you could have chosen (that) or the only one available to you (, which)?

Lines 130-131: ...within a 40 km radius of Hyytiala.

Line 134: ...Leino et al. (2019), and Lampilahti et al. . .

Line 138: ...used. Bragg scatter occurs in areas where there are. . .

Line 142: ..."leads to eddies with" or "generates eddies with"

Line 143: Change semicolons to commas and add "and" before Tanamachi

Line 148: Unsure what you mean by smaller numerous. Fewer?

Line 154: Remove comma after water cloud.

Line 182: Add comma after Bragg scatter.

Line 184: ...of values in the Bragg histogram...

Line 197: ...can be either Bragg or insects...

Line 198: ...area corresponds to the entrainment...

Line 203: Should be (Fig. 1c and 1d)?

Line 206: Either "quite well-seen" or "easily seen" or "seen quite well"

Line 212: Would you call the insect process a filter? Perhaps ...insects filter from...

Lines 218-219: ...shown in Fig. 6 using...

Line 220: Add comma after Halo lidar

Line 221: Remove "so" after not and before many insects.

Line 227: Interestingly, this zone is still present...

Line 230: ...temperature during the descent...

Line 236: The transition between Cessna location and noticeable is a bit awkward. What do you intend to say?

Line 237: ...flight days are need to make a conclusion on the depth...

Line 240: In the presence of low level clouds or precipitation,¡space¿the...

Lines 248-250: The BL started to develop after the rain had passed, which can be seen in the radars' echoes from 500 m at 06.00 UTC. The BL derived from Ka-band, similarly to the first case, shows more insects and at a bit higher altitude¡no s¿ compared to the other radars.

Line 254: Can you surmise about why that layer of Bragg scatter may have developed? It seems to me that a residual layer is unlikely given antecedent precipitation, but it is curious why that would show up. If possible it would be helpful to add an additional sentence to that paragraph indicating what it could be, or at the very least ruling out the possibility of a residual layer.

*We do not have a good hypothesis of why this Bragg layer appeared, but we agree that it is not a residual layer and we added to the text (line 254):"**There is also an interesting part of Bragg scatter in between precipitation at 04.00 UTC, which cannot be a residual layer due to heavy precipitation during the night."***

Line 255: Change "between" to "with"

Line 283: Upper left "corner" is a bit too strong of a word, which led my eyes astray from the points you were trying to make. Perhaps "top left quadrant" is a better descriptor.

Line 307: ...Bragg scatter area corresponds to the CBLH.

Lines 309-310: ...upper edge was "up to" 20 meter higher... Not clear what you're trying to communicate here, but a bit more clarification would be helpful.

Figure 2: Change "that" to "which" in line 2 of the caption. Change end of sentence to "higher values for insects because of their non-spherical shape".

Figure 5: Note the date somewhere in the caption.

Figure 8: Add spaces after "b)" and "d)".

Figure 9: Add spaces after "a)" and "b)".

Figures 10, 11: Suggest different colors for ERA and Doppler lidar lines. Grey and black are a bit hard to differentiate.

---

## Referee Report (RR1)

**Evaluation of convective boundary layer height estimates using radars operating at different frequency bands**
*Anna Franck, Dmitri Moisseev, Ville Vakkari, Matti Leskinen, Janne Lampilahti, Veli-Matti Kerminen, and Ewan O'Connor*

By: Anonymous Referee #2

*Second Round of Revisions*

General Comments
The authors are to be commended for thorough handling of my comments and suggestions. I believe the manuscript presents a compelling result and is communicated in a concise, accurate, and effective way.

Specific Comments
My only remaining suggestion is as follows: When reading through the manuscript again in its current form, the authors, perhaps inadvertently, lead the reader to believe that the ERA5 CBLH estimates are the "truth" to which the observational CBLH estimates are compared. Since the ERA5 BLH parameter is imperfect, the authors cannot make absolute statements about the over- or underestimation of radar-derived CBLH. Method-relative comparisons should be used instead. To address this subtlety, I recommend the following:

1. Include an estimate of ERA5 BLH parameter error bars (if knowable) in Section 2.3.
2. Remove all absolute statements about radar-derived CBLH accuracy.
   a. Line 279: Change "underestimated" to "slightly lower than the ERA5 and Halo Doppler lidar CBLH estimates."
   b. Line 281: Change "good" to "comparable to that of the ERA5 and Halo Doppler lidar CBL estimates"
   c. Line 303: Change "verified with" to "compared to".

Technical Corrections
Line 283                  Change "Halo Doppler radar" to "Halo Doppler lidar".

---

## Author Response (AR2)

**Author's reply to the Anonymous Referees, revised submission**

We thank the Referees once again for reviewing our manuscript.

**Author's reply to the Anonymous Referee #1**

*We have implemented all technical corrections suggested by the Referee:*

- On line 210, add a space between the m and s in the unit m s$^{-1}$. Add the unit to the second time the value is mentioned (i.e., write "0.5 m s$^{-1}$", instead of "0.5$^{-1}$").

- Line 265: change to "Underestimation of the CBLH from the C-band data"

- Line 266: change to "while underestimation from the W-band data"

- Line 266: change to "flaws in the algorithm"

- Line 288: change "K-band" to "Ka-band"

- Line 289: change to "and are chosen to be"

- Line 290: change to "in the millimeter wavelength radars"

- Line 313: change to "in some cases a time lag"

- Line 314: change to "the temperature threshold for insect flight", the word 'comfort' is a bit confusing here and so I suggest you remove it

- Line 318: change to "when precipitation would have to cease"

- Line 319: change to "where different insect flight behaviour"

- Line 331: change to "of this type of radar is lower"

**Author's reply to the Anonymous Referee #2**

*Specific Comments*

1. Include an estimate of ERA5 BLH parameter error bars (if knowable) in Section 2.3.

**ERA5 BLH parameter error is not known, therefore we could not include it.**

**We have implemented all technical corrections and specific comments suggested by the Referee:**

2. Remove all absolute statements about radar-derived CBLH accuracy.

a. Line 279: Change "underestimated" to "slightly lower than the ERA5 and Halo Doppler lidar CBLH estimates."

b. Line 281: Change "good" to "comparable to that of the ERA5 and Halo Doppler lidar CBL estimates"

c. Line 303: Change "verified with" to "compared to".

*Technical Corrections*

Line 283 Change "Halo Doppler radar" to "Halo Doppler lidar".